# Ataluren binds to multiple protein synthesis apparatus sites and competitively inhibits release factor-dependent termination

Shijie Huang[1,3,4], Arpan Bhattacharya [1,2,4], Mikel D. Ghelfi[1], Hong Li[1], Clark Fritsch [2], David M. Chenoweth[1], Yale E. Goldman[2,5] & Barry S. Cooperman [1,5✉]

Genetic diseases are often caused by nonsense mutations, but only one TRID (translation readthrough inducing drug), ataluren, has been approved for clinical use. Ataluren inhibits release factor complex (RFC) termination activity, while not affecting productive binding of near-cognate ternary complex (TC, aa-tRNA.eEF1A.GTP). Here we use photoaffinity labeling to identify two sites of ataluren binding within rRNA, proximal to the decoding center (DC) and the peptidyl transfer center (PTC) of the ribosome, which are directly responsible for ataluren inhibition of termination activity. A third site, within the RFC, has as yet unclear functional consequences. Using single molecule and ensemble fluorescence assays we also demonstrate that termination proceeds via rapid RFC-dependent hydrolysis of peptidyl-tRNA followed by slow release of peptide and tRNA from the ribosome. Ataluren is an apparent competitive inhibitor of productive RFC binding, acting at or before the hydrolysis step. We propose that designing more potent TRIDs which retain ataluren's low toxicity should target areas of the RFC binding site proximal to the DC and PTC which do not overlap the TC binding site.

[1] Department of Chemistry, University of Pennsylvania, Philadelphia, PA 19104, USA. [2] Department of Physiology, Perelman School of Medicine, University of Pennsylvania, Philadelphia, PA 19104, USA. [3] Present address: GSK, 14200 Shady Grove Rd, Rockville, MD 20850, USA. [4] These authors contributed equally: Shijie Huang, Arpan Bhattacharya. [5] These authors jointly supervised this work: Yale E. Goldman, Barry S. Cooperman. ✉email: cooprman@upenn.edu

Nonsense mutations, leading to the introduction of a UGA, UAG, or UAA premature stop codon (PSC) within open reading frames, result in premature termination of protein synthesis, giving rise to many human disorders[1,2]. More than 11% of known human inherited genetic diseases, including variants of cystic fibrosis, Duchenne muscular dystrophy (DMD), and some forms of cancers are caused by nonsense mutations[3]. Several small molecules, denoted translation readthrough inducing drugs (TRIDs)[4], are known to promote at least partial readthrough of PSCs, leading to insertion of an amino acid at the PSC site[5,6]. Currently, the best characterized TRIDs are ataluren (Translarna), a hydrophobic substituted oxadiazole identified as a TRID by PTC Therapeutics Corp., using high throughput screening[7,8], and the highly polar class of aminoglycoside antibiotics (AGs)[9]. As yet, only ataluren has been approved for clinical use. This approval, by the European Medicines Agency, is currently specified for treatment of patients with nonsense-mediated DMD. In addition, ataluren has been shown to be effective in restoring the expression of more than 20 disease-associated genes in vitro[10] and in vivo[11–15], and clinical trials are underway for treatment of other diseases (aniridia, NCT02647359; epilepsy, NCT0275826; and colorectal and endometrium cancers, NCT04014530). On the other hand, ataluren has not yet received clinical approval from the US Food and Drug Administration, and, as recently reviewed[6,16], has been shown to be ineffective in stimulating readthrough in some cellular studies and in achieving favorable outcomes in clinical studies of several other nonsense-mediated diseases. The clinical utility of aminoglycosides has been restricted, in part, by their toxicities[17]. Nevertheless, a Phase 2 clinical trial has recently begun (cystic fibrosis, NCT04135495) with a molecularly engineered AG, ELX-02 (formerly known as NB124), having much reduced toxicity[18].

Efforts to use a rational approach to improve upon the limited clinical success of ataluren have been hampered by the lack of detailed knowledge of both its mechanism of action and its site or sites of interaction with the cell's protein synthesis machinery. Recent cellular studies have provided important details of its relative activities towards the three different stop codons and a determination of which amino acids are inserted into nascent protein chains at PSCs in its presence[19,20]. However, these cell-based assays left unanswered the questions of which specific process or processes within polypeptide elongation or termination does ataluren affect, and where does it interact? To address these questions directly, we have employed a highly purified, eukaryotic cell-free protein synthesis system, denoted PURE-LITE. PURE-LITE is based on the ability of the intergenic internal ribosome entry site (IRES) of Cricket Paralysis Virus (CrPV-IRES) to form a complex with ribosomes capable of initiating polypeptide synthesis in the complete absence of native cytoplasmic initiation factors. In the work described below, PURE-LITE consisted of the following components: 80 S ribosomes purified from shrimp cysts, translation factors eEF1A, eEF2, eRF1, and eRF3 from Saccharomyces cerevisiae, and aminoacylated tRNAs prepared from S. cerevisiae and Escherichia coli. As described earlier, despite the heterologous nature of this system, we demonstrated that PURE-LITE could recapitulate some of the effects on readthrough seen in live cell assays of TRIDs, including ataluren, other hydrophobic heterocycles, dubbed ataluren-like, and aminoglycosides[21]. This result was not unexpected, given that eukaryotic translation factors have very strongly conserved structures[22–24] and charged tRNAs from one species typically form functional complexes with both eEF1A and ribosomes from a different species[25,26]. We have further demonstrated that ataluren and the aminoglycoside G418 (geneticin) stimulate readthrough by orthogonal mechanisms. Ataluren stimulation derives exclusively from its ability to inhibit release factor complex (RFC, eRF1.eRF3.GTP) activity, whereas G418, binding to its high affinity site on the ribosome, increases functional near-cognate tRNA mispairing, with little if any effect on RFC activity[27]. The low toxicity of ataluren, in contrast to the much higher toxicity of aminoglycosides which has thus far prevented their clinical use, emphasizes the desirability of developing new TRIDs targeting release factor activity.

Here we identify sites of interaction of ataluren within the protein synthesis apparatus by photoaffinity labeling (PAL) experiments utilizing a [3H]-labeled photolabile congener of ataluren, [3-(5-(4-azidophenyl)-1,2,4-oxadiazol-3-yl) benzoic acid], referred to as AzAt throughout the text (Fig. 1a). The targets of the PAL probe are the release factor complex, eRF1.eRF3.GDPNP, in which the non-hydrolyzable analog GDPNP substitutes for GTP, the Stop-POST5 complex depicted in Fig. 1b, containing FKVRQ-tRNA^Gln in the P-site adjacent to the UGA stop codon in the A-site, and the 80S.IRES complex, which lacks FKVRQ-tRNA^Gln. Stop-POST5 is an example of a pretermination complex, which we have previously used to characterize ataluren inhibition of RFC-dependent termination[27]. Using the PAL approach we identify three sites of ataluren binding within the PURE-LITE system, two within rRNA and one within eRF1. We also use Stop-POST5 as a substrate in the termination reaction, the kinetics of which we examine in detail by single molecule and ensemble approaches using fluorescent derivatives of FKVRQ-tRNA^Gln. These experiments demonstrate that ataluren acts as a competitive inhibitor of RFC through its binding to multiple sites within the protein synthesis apparatus. They further show that tRNA and peptide release from the ribosome proceed slowly following the much more rapid cleavage of the peptidyl-tRNA bond, and that ataluren inhibition is exerted only at or before the cleavage step. Both the structural and kinetic results indicate that ataluren inhibits release factor activity via a cooperative binding mechanism and lead to a suggestion of which areas of the ribosome could be targeted in the ongoing efforts to develop more effective TRIDs.

## Results

**Photoaffinity Labeling (PAL) Experiments**. AzAt mimics ataluren in stimulating readthrough of a premature stop codon (Supplementary Fig. 1) and in inhibiting eRF1/eRF3-catalyzed termination at a premature stop codon[27], showing a lower EC$_{50}$ than ataluren in both assays. Photochemical experiments on aryl azides have demonstrated that photolysis of a phenyl azide

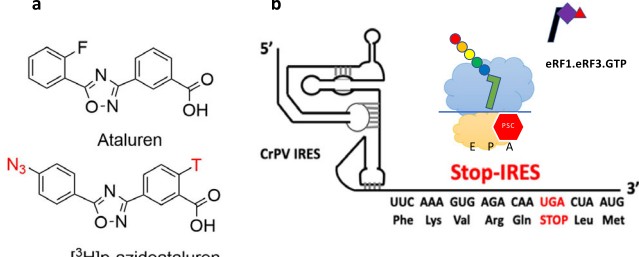

**Fig. 1 Targets of azidoataluren photoincorporation. a** Structures of ataluren and [3H]-AzAt. T refers to tritium. N$_3$ is the azide which confers photolability on azidoataluren. **b** Stop-IRES mRNA encoding FKVRQStopLM. The cartoon depicts the Stop-POST5 complex containing FKVRQ-tRNA^Gln bound in the P-site adjacent to an empty A-site containing the UGA stop codon, and the incoming eRF1.eRF3.GTP complex which catalyzes cleavage of the ester bond linking FKVRQ to tRNA^Gln after binding to the A-site.

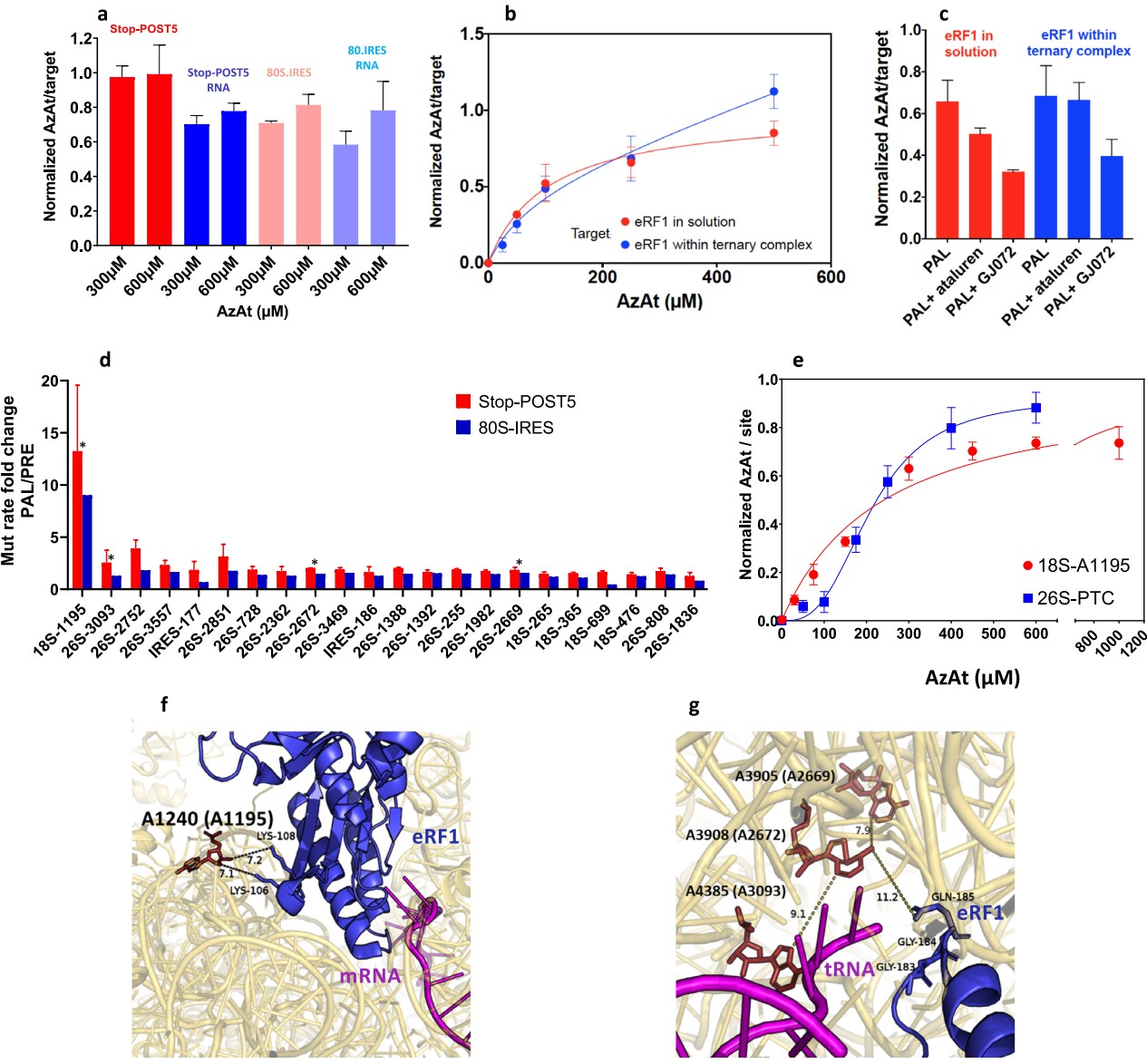

**Fig. 2 AzAt Photoaffinity Labeling. a** AzAt photoincorporation into Stop-POST5, the RNA fraction of Stop-POST5, 80S.IRES, and the RNA fraction of 80S.IRES. All of the labeling stoichiometries are normalized to that of Stop-POST5 at 600 μM AzAt, which was equal to 1.2/Stop-POST5. **b** AzAt photoincorporation into eRF1 both alone and complexed with eRF3.GDPNP. The values are normalized to the saturation labeling of isolated eRF1. **c** Inhibition of AzAt (250 μM) photoincorporation by the addition of either ataluren (1000 μM) or GJ072 (150 μM). **d** The mutation rate fold change for PAL vs. PRE samples for Stop-POST5 (red) and 80S.IRES complexes (blue) for the 22 sites most pertinent for ataluren function, and the asterisks indicates sites of particular interest (see text). **e** Saturation curves for photoincorporation into 18S-A1195, as measured by photoincorporation into Fragment I, vs. the sum of the photoincorporations into 26S A3093, A2669, and A2672, as measured by photoincorporation into Fragments II and III. **f** Location of A1195 within the 40S subunit containing bound eRF1. **g** Locations of A2669, A2672, and A3093 within the 60S subunit containing bound eRF1. All of the error bars in this Figure represent average deviations for $n = 2$ independent determinations, with the exception of the eRF1 labeling within the ternary complex in b, for which n=3. Source data for (**a**–**e**) are provided as a Source Data file.

rapidly forms (in ≤20 ns) either a ketenimine or azepinone intermediate, which reacts with nucleophiles in approximately 100 μs, increasing the probability that photoincorporation of photolyzed AzAt will occur in proximity of its binding site. In addition, the reactive intermediates are rapidly quenched by an amine-containing buffer[28], such as the Tris buffer employed in our studies, minimizing photoincoporation into a biological target resulting from azide photolysis occurring in solution. We carried out PAL experiments with [³H]-AzAt in order to identify the binding sites of ataluren within the protein synthesis apparatus of the PURE-LITE system. We separately performed AzAt-PAL experiments on the Stop-POST5 complex, the 80S.IRES

complex, and eRF1, both alone and within the eRF3.GDPNP ternary complex, giving the photoincorporation (PI) values shown in Fig. 2a–c. Here it is important to point out that our preparation of Stop-POST5 had a stoichiometry of 0.40 ± 0.05 PheLysValArgGln-tRNA$^{Gln}$/ribosome, with the remaining fraction of the ribosome population consisting of the 80S.IRES complex. As a result, while PI into the 80S.IRES complex and eRF1 could be measured directly, the PI values for the Stop-POST5 complex are values calculated using Eq. (1), which corrects for the fraction of 80S.IRES complex present in the Stop-POST5 preparation. The measured, uncorrected values for the Stop-POST5 complex as a function of AzAt concentration are

shown in Supplementary Fig. 2.

$$PI(Stop - POST5)_{corrtd} = 2.5 \{PI(Stop - POST5)_{measured} \\ - 0.6\, PI(80S.IRES)_{measured}\} \quad (1)$$

Photoincorporation is about 30% higher for Stop-POST5 than for the 80S.IRES complex and, for both targets, most of the photo-incorporation (≥70%) is found in the RNA fraction, which comprises all six RNA species (26S rRNA, 18S rRNA, 5.8S rRNA, 5S rRNA, mRNA, and tRNA) but lacks r-protein. As determined by urea-PAGE analysis (Supplementary Fig. 3a) >90% of the RNA photoincorporation is found within the 18S and 26S rRNA, with the remainder distributed among the smaller RNAs (Supplementary Table 1).

We also measured AzAt photoincorporation into either eRF1 or eRF3.GDPNP alone, or into the eRF1.eRF3.GDPNP ternary complex (Fig. 2b). Photoincorporation into isolated eRF1 as a function of AzAt concentration results in a simple hyperbolic saturation curve with an $EC_{50}$ of $110 \pm 20\, \mu M$. The corresponding experiment performed on eRF1 within the eRF1.eRF3.GDPNP ternary complex shows biphasic behavior, consistent with a tight binding site and a much weaker site. The tight site photo-incorporation proceeds with an $EC_{50}$ value of $90 \pm 20\, \mu M$, indistinguishable from that found with isolated eRF1. Importantly, both ataluren and GJ072, an ataluren-like TRID[21,27], substantially inhibit AzAt photoincorporation into isolated eRF1, although the inhibition is less marked within the ternary complex (Fig. 2c). Although AzAt also photoincorporates into isolated eRF3, such photoincorporation is due to either a very weak site binding or to reaction with AzAt photolyzed in solution before binding, since it increases in a strictly linearly fashion up to an AzAt concentration of $500\, \mu M$. Photoincorporation into eRF3 is strongly inhibited within the ternary complex (Supplementary Fig. 4), perhaps due to surface residues in isolated eRF3 becoming buried within the ternary complex.

**Identifying RNA sites photolabeled by AzAt via nitrene photochemistry**. The dominance of AzAt photoincorporation into the RNA fraction of both Stop-POST5 and 80S.IRES complexes led us to use LASER-seq[29] to identify the sites of AzAt photo-incorporation into RNA. In this method, modified nucleotides show up as point mutations introduced during reverse transcription. We carried out LASER-Seq on four types of samples, denoted PAL, PRE, UV, and NUL. PAL samples were prepared by photoaffinity labeling of complexes with $300\, \mu M$ AzAt, a concentration close to saturating for photoincorporation (Fig. 2a). PRE samples were prepared using separately prephotolyzed AzAt in Buffer 4 (Methods) which was re-irradiated in the presence of complexes. UV and NUL samples were prepared in the absence of AzAt by either subjecting complexes to uv irradiation (UV) or just analyzing complexes directly (NUL).

LASER-Seq read coverages were virtually identical for all samples (Supplementary Fig. 5) with, in each case, over 97% of the nucleotides giving at least 10,000 reads. Notable exceptions are tRNA$^{Gln}$, 26S rRNA 1763-1801 and 18S rRNA 954-956 which gave very low reads, likely due to especially strong secondary structure, and were excluded from downstream analysis. The raw RNA-seq results are presented in Supplementary Data 1. We found very high mutation rates (28–100%) in 10 nucleotides in all samples (Supplementary Table 2). Three of these nucleotides have previously been reported to be modified at the conserved sites in *S. cerevisiae* {25S-2870, m$^5$C; 18S-1191, m$^1$acp$^3\Psi$;[30] 25S-A645, m$^1$A[31]} two of which were also reported in human rRNA {28S-A1314, Am; 18S-U1248, m$^1$acp$^3\Psi$[32]}. To our knowledge, the remaining seven natively modified sites have not been reported previously. In addition, relatively high mutation rates (1.8–9.9%)

were found in 14 nucleotides in the PAL, PRE, and UV samples (Supplementary Table 3) but not in the NUL samples, and likely include sites of uv-induced crosslinking[33].

We first identified the sites of modification arising exclusively via nitrene formation on irradiation of AzAt by the mutation rate fold change of PAL vs. PRE samples, a procedure which corrects for mutation rates arising from either noncovalent binding, covalent photoincorporation of pre-photolyzed AzAt, or uv-induced mutations not related to AzAt (Supplementary Table 3). The results obtained for the ≥6006 nucleotides present in these two complexes are presented in Supplementary Fig. 6. In examining these data sets to identify potentially interesting sites of AzAt labeling of the Stop-POST5 and 80S.IRES complexes, we initially imposed the constraints that (i) the fold-increase (FI) was ≥ 1.5; (ii) there were ≥100 reads for each mutated site; and (iii) the results were statistically significant (Z-factor > 0 and *p*-value < 0.05). These constraints narrowed the number of potentially interesting sites to the 22 identified in Fig. 2d. Strikingly, nucleotide 18S-A1195 has by far the highest fold-increase in both complexes, suggesting that it arises from a site present in both.

The large number of nucleotides of potential interest identified in Fig. 2d raised the question of which are most likely to account for ataluren inhibition of release factor activity. To address this question we first used the structure of the fully accommodated eRF1 bound to a termination complex of the rabbit ribosome (PDB: 5LZU) as a reference to identify nucleotides labeled by AzAt which are the most proximal to eRF1. Most of the nucleotides in Fig. 2d have distances of closest approach to eRF1 ≥ 40 Å (Supplementary Fig. 7) and are located at peripheral and flexible loop regions with low evolutionary conservation among eukaryotic species, suggesting that they are less likely to arise from an AzAt site which interferes with eRF1 binding. In contrast, three labeled nucleotides, 18S-A1195, 26S-A3093, and 26S-A2672, each fall within 10 Å of eRF1, and a fourth, 26S-A2669, falls with 16 Å (Supplementary Fig. 7). These four nucleotides, all of which are in highly conserved and functional regions of rRNA (Supplementary Fig. 8), are indicated with asterisks in Fig. 2d. 18S-A1195 is proximal to the segment of eRF1 which binds to the termination codon during translation termination (Fig. 2f), is part of the conserved helix 31 loop region, and forms a polar pocket with Tyr123, Lys124, and Lys100 of r-protein uS19 (Supplementary Fig. 9), a component of the decoding center which is functionally involved in multiple states of translation elongation[34]. As shown in Fig. 2g, 26S-A3093, 26S-A2672, and 26S-A2669 are clustered within the peptidyl transferase center (PTC), in the vicinity of the catalytically important GGQ motif of eRF1, and extending toward the peptide exit tunnel. In addition, two CrPV-IRES nucleotides, U177 and G186, have significant fold-increases, and are clustered within the stem-loop region of the CrPV-IRES pseudoknot PKI (Supplementary Fig 10), which mimics the anti-codon stem-loop of the initiator tRNA during translation initiation. As they are not present in PDB:5LZU it is unclear whether they are proximal to the eRF1 binding site in the POST5 complex, since it is likely that the CrPV IRES moves away from the tRNA binding sites as the nascent peptide is elongated. Because of this uncertainty, we included these two nucleotides along with the proximal four mentioned above in our further efforts to identify functionally important ataluren binding sites.

**Identifying functionally important sites photo labeled by AzAt**. We employed an RNAse H fragment assay to test which of the six nucleotides selected above correspond to ataluren binding sites relevant for RFC activity. In this assay, we used oligonucleotide

**Table 1 Photoincorporation yields (mole%) into RNAse H Fragments[a].**

| Target | Experiment | Competing Ligand[b] | Mutated site(s)/RNAse Fragment | | | |
|---|---|---|---|---|---|---|
| | | | 18S A1195/I: 18S 1151–1206 | 26S-A3093/II: 26S 3069–3113 | 26S-A2669;A2672/III: 26S 2638–2702 | CrPV-IRES U177;G186 IV: 156–221 |
| Stop-POST5 | PAL, 300 μM AzAt | - | *1.55 ± 0.20* | 0.40 ± 0.01 | 1.07 ± 0.14 | 3.2 ± 0.8 |
| | | eRF1/eRF3/GDPNP | 0.07 ± 0.03 | 0.18 ± 0.10 | 0.19 ± 0.02 | 2.9 ± 0.6 |
| | | eRF1 | 0.19 ± 0.07 | 0.18 ± 0.01 | 0.20 ± 0.01 | - |
| | PAL, 30 μM AzAt | - | 0.26 ± 0.03 | 0.03 ± 0.01 | 0.13 ± 0.04 | - |
| | | Ataluren, 500 μM | 0.06 ± 0.01 | 0.06 ± 0.01 | 0.10 ± 0.04 | – |
| | | GJ072, 150 μM | 0.10 ± 0.01 | 0.07 ± 0.01 | 0.07 ± 0.01 | – |
| | PRE, 300 μM AzAt | - | 0.012 ± 0.003 | 0.04 ± 0.01 | 0.03 ± 0.02 | – |
| 80S-IRES | PAL, 300 μM AzAt | - | *1.03 ± 0.33* | *0.50 ± 0.09* | *1.01 ± 0.12* | – |
| | | eRF1/eRF3/GDPNP | 0.56 ± 0.06 | 0.55 ± 0.12 | 0.94 ± 0.23 | – |

[a]Error ranges are ± average deviations, $n = 2$ or 3 (italicized) independent determinations; [b]Added concentrations: eRF1, 2 μM; eRF3, 4 μM; GDPNP, 1 mM

hybridization, RNase H digestion and polyacrylamide gel electrophoresis (PAGE) to determine the stoichiometry of [³H]-AzAt photo incorporation, measured using 300 μM AzAt, paralleling the LASER-Seq PAL experiments, into the three rRNA fragments and one CrPV fragment that together include the six nucleotides of potential interest: Fragment I, 18S-A1195; Fragment II, 26S-A3093; Fragment III, 26S-A2669 and 26S-A2672; and Fragment IV, CrPV U177 and G186 (Table 1). In this assay, the background in the PRE sample is negligible.

We next asked whether addition of eRF1.eRF3.GDPNP complex inhibits such photoincorporation, consistent with the apparent competitive inhibition of RFC activity by ataluren, as demonstrated below. The results of these experiments, summarized in Table 1, clearly demonstrate that eRF1.eRF3.GDPNP strongly inhibits AzAt photoincorporation into Fragments I–III, while having essentially no effect on photoincorporation into Fragment IV. These results provide a clear suggestion that photoincorporation within Fragments I–III takes place within the eRF1.eRF3.GDPNP binding site, while photoincorporation into Fragment IV does not. As a result, our subsequent experiments focused on determining the effects on AzAt photoincorporation into Fragments I–III of various modifications in experimental protocol. Further photoincorporation results into Fragment IV, and for several other RNA sites showing somewhat high AzAt photoincorporations but which were not strongly inhibited by eRF1.eRF3.GDPNP, are presented in Supplementary Information (Supplementary Table 1 and Supplementary Notes).

Somewhat surprisingly in view of published cryoEM structures[35], added eRF1 alone has very similar effects on photoincorporation into Fragments I–III as the eRF1.eRF3.GDPNP complex, a point we return to in the Discussion. Photoincorporation into Fragments I–III derived from 80 S.IRES is roughly similar to what we find for Stop-POST5, but the effects of added eRF1.eRF3.GDPNP complex are quite different, with no measured inhibition of photoincorporation into Fragments II and III and only partial inhibition of photoincorporation into Fragment I. These results reflect the expected weaker binding of the RFC to the ribosome in the absence of a stop codon at the A-site, and also imply that such weaker binding is not accompanied by a conformational change that results in eRF1 interaction with the PTC.

Next we determined whether added ataluren or the ataluren-like TRID GJ072 inhibited AzAt photoincorporation into Fragments I – III (Table 1) of the Stop-POST5 complex. These experiments were performed at a much lower AzAt concentration than those mentioned above (30 μM vs. 300 μM), because we expected that inhibition would be incomplete at high AzAt concentration, based on the relative $EC_{50}$ values of ataluren > AzAt ≈ GJ072 as termination inhibitors[27] and the

limited solubility of GJ072 in aqueous medium. Photoincorporation into Fragment I, reflecting 18S A1195 labeling, is strongly inhibited by ataluren and GJ072. Taken together with the proximity of 18S A1195 to a region of eRF1 that interacts with mRNA in the decoding center of the small subunit (Fig. 2f), the results in Table 1 provide strong evidence that 18S A1195 is labeled from a functionally important ataluren binding site which inhibits both eRF1.eRF3.GDPNP and eRF1 binding. However, photoincorporation into Fragment III, reflecting A2669/A2672 photolabeling, is only weakly inhibited by ataluren and GJ072, and photoincorporation into Fragment II, reflecting A3093 photolabeling, is actually stimulated. These results could indicate that photolabeling of the PTC sites does not proceed from a true ataluren binding site, but an alternative interpretation is that the results arise from two opposite effects: a direct competition by ataluren and GJ072 with AzAt at the PTC and an allosteric stimulation of AzAt binding at the PTC by ataluren or GJ072 binding elsewhere within the ribosome. To resolve this uncertainty we determined the AzAt concentration dependence of photoincorporation into these PTC sites (Fragments II and III) and into A1195 (Fragment I). We found that photoincorporation into Fragment I can be fit to a simple hyperbolic binding isotherm with an $EC_{50}$ of $240 ± 10$ μM (Fig. 2e), but that photoincorporation into Fragments II and III both show clear evidence of cooperativity, with saturation curves which are quite similar to one another (Supplementary Fig. 11). This latter result, and the mutual proximities of 26S-A3093, 28S-A2672, and 28S-2669 (Fig. 2g), suggest that photoincorporation into these three nucleotides occurs from a single binding site within the PTC. Accordingly, in Fig. 2e we compare the normalized values of the combined saturation curve for Fragments II and III with the normalized values for the saturation curve of Fragment I, showing the clear difference in the shape of the curves. Fitting the saturation curve for Fragments II and III to Eq. (2) yielded a $K_A$ equal to $200 ± 40$ μM and a Hill $n$ of $3.0 ± 0.3$.

$$PI = \frac{PI_{max}[AzAt]^n}{\left(K_A{}^n + [AzAt]^n\right)} \quad (2)$$

**Ataluren is an apparent competitive inhibitor of RFC-dependent termination of polypeptide synthesis.** We used two different fluorescent assays to quantify ataluren inhibition of termination. As described in Table 2, smTIRF assays employed three doubly-labeled forms of the Stop-POST5 complex, Atto(pep)-Cy(tRNA)-Stop-POST5, Atto(rbsm)-Cy(tRNA)-Stop-POST5, and Atto(pep)-Cy(rbsm)-Stop-POST5. Ensemble fluorescence anisotropy

**Table 2 Sites of fluorophore incorporation within fluorescent Stop-POST5 complexes.**

| Stop-POST5 | Lys within FKVRN | tRNA$^{Gln}$ | Ribosome[a] |
|---|---|---|---|
| Atto(pep) | Atto647 | unlabeled | unlabeled |
| Atto(pep)-Cy(tRNA) | Atto647 | Cy3 | unlabeled |
| Atto(rbsm)-Cy(tRNA) | unlabeled | Cy3 | Atto647N |
| Atto(pep)-Cy(rbsm) | Atto647 | unlabeled | Cy3 |

[a]ribosomes were labeled to a total stoichiometry of ~1/ribosome spread over Lys residues within r-proteins

studies employed the singly-labeled form, Atto(pep)-Stop-POST5. In Atto(pep)-Stop-POST5, Atto(pep)-Cy(tRNA)-Stop-POST5, and Atto(pep)-Cy(rbsm)-Stop-POST5, the lysine epsilon amino group in the pentapeptide is derivatized with Atto-647. In Atto(pep)-Cy(tRNA)-Stop-POST5 and Atto(rbsm)-Cy(tRNA)-Stop-POST5, a dihydrouridine residue of tRNA$^{Gln}$ is derivatized with Cy3. In Atto(rbsm)-Cy(tRNA)-Stop-POST5, the ribosome is labeled with Atto-647N. In Atto(pep)-Cy(rbsm)-Stop-POST5, the ribosome is labeled with Cy3.

**smTIRF results using Atto(pep)-Cy(tRNA)-Stop-POST5 attached to surface-immobilized ribosomes.** Although the distance between the two fluorophores is not close enough to produce FRET, fluorescent pairs could be identified by co-localization on the slide under alternating 532 nm and 640 nm TIRF illumination. This permits near-simultaneous monitoring of the dynamics of peptide and P-site tRNA release from the Stop-POST5 complex following RFC binding. To reduce photobleaching of each fluorophore, we recorded time-lapse movies, with bursts of 20 or 50 frames at 100 ms frame intervals interleaved by 24 s intervals with the lasers shuttered. Total recording time for each trial was 22.7 min. Figure 3a, c show typical frame by frame recordings indicating a loss of signal for both labeled peptide and labeled tRNA after the addition of the RFC to Atto(pep)-Cy(tRNA)-Stop-POST5. The corresponding results plotted in actual time are shown in Fig. 3b, d. Either peptide (a) or tRNA (b) can dissociate first, in approximately equal proportions. Photobleaching of Cy3-labeled tRNA and Atto647-labeled peptide in the absence of eRF1/eRF3 was much slower (Fig. 3e, f). Under the time lapse illumination employed, mean lifetimes for Atto647-peptide and Cy3-tRNA at 0.32 µM eRF1 and 2 µM eRF3 were each 2.45 min. By comparison, in the absence of RFC, the mean lifetimes under time-lapse illumination before Atto647 and Cy3 disappearance were 6.8 min and 4.9 min, respectively, which were mainly due to photobleaching. Since the single-step decrease of fluorescence intensity is due either to photobleaching or to dissociation of peptide or tRNA from the ribosome, we calculate the RFC-dependent rate constant for peptide or tRNA dissociation, $k_{RF}$, as equal to $1/<T_{obs}> - 1/<T_{pb}>$, where $<T_{obs}>$ is the mean observed lifetime and $<T_{pb}>$ is the mean time until signal loss in the absence of RFs.

We next determined $k_{RF}$ values for peptide and tRNA dissociation as a function of both RFC and ataluren concentrations. Figure 4a, b shows examples of cumulative distributions of peptide and tRNA dissociation times at 320 nM RFC, 32 nM RFC, and 32 nM RFC plus 1 mM ataluren. Fitting these results to single exponentials and subtracting the photobleaching rate gave average $k_{RF}$ values. In the absence of ataluren, values of $k_{RF}$ as a function of RFC concentration fit well to the Michaelis-Menten equation, yielding values of $V_{max} = \sim0.27 \pm 0.02$ min$^{-1}$ peptide and $0.20 \pm 0.02$ min$^{-1}$ for

peptide and tRNA dissociation, respectively, and an $EC_{50} = \sim0.02$ µM for each (Fig. 4c, d, red curves). Addition of ataluren at concentrations of 200, 500, and 1000 µM (Fig. 4c, d) raised $EC_{50}$ values significantly, reaching ~0.1 µM at 1000 µM, whereas $V_{max}$ values at high RFC concentration were little affected, a strong indication that ataluren acts as an apparent competitive inhibitor of RFC. Separate control experiments performed in the absence of ataluren with Atto(rbsm)-Cy(tRNA)-Stop-POST5 and Atto(pep)-Cy(rbsm)-Stop-POST5 at a saturating concentration of RFC (0.65 µM eRF1, 1.3 µM eRF3) gave rate constants, $0.20 \pm 0.03$ min$^{1}$, and $0.26 \pm 0.03$ min$^{-1}$ for tRNA and peptide dissociation, respectively, indistinguishable from those measured for Atto(pep)-Cy(tRNA)-Stop-POST5, showing that labeled peptide did not inhibit tRNA dissociation and labeled tRNA did not inhibit peptide dissociation.

**Ensemble fluorescence anisotropy results using Atto(pep)-Stop-POST5.** The fluorescence anisotropy of the pentapeptidyl moiety of Atto(pep)-Stop-POST5 is quite high (0.22), and decreases following addition of RFC, providing a convenient measure of the rate of pentapeptide release from the ribosome. Sample results show the time dependence of anisotropy decrease as a function of RFC concentration (Fig. 4e), and, at fixed RFC, of ataluren concentration (Fig. 4f). Each such time-dependent anisotropy decrease could be fit to a single exponential, giving the collected rate constants presented in Fig. 4g, in which the rate dependence as a function of RFC concentration is measured at 0, 200, 500, and 1000 µM ataluren. In the absence of ataluren, we obtained $EC_{50}$ ($0.029 \pm 0.002$ µM) and $V_{max}$ ($0.31 \pm 0.02$ s$^{-1}$) values for pentapeptide release similar to those we measured by smTIRF. In addition, as with the smTIRF results, $V_{max}$ values were unaffected by added ataluren but $EC_{50}$ values increase, reaching ~0.06 µM at 1000 µM ataluren.

**Ataluren inhibition of RFC activity is cooperative.** The observed dissociation rate constants for pentapeptide release, measured by both smTIRF and fluorescence anisotropy, show very similar sigmoidal dependences on added ataluren concentration (Fig. 4h), giving a Hill n value of $3.0 \pm 0.6$ and a $K_A$ of $250 \pm 20$ µM (see Eq. 3). Similar results were obtained for tRNA release measured by smTIRF (Hill $n = 2.7 \pm 0.4$, $K_A = 180 \pm 10$ µM).

**Further control experiments.** The rates of complete release of the two products, ~0.2–0.3 min$^{-1}$, observed here at 24 °C–25 °C (Fig. 4c, d, g, h) are ~20–30 times slower than the rate constants of 5–10 min$^{-1}$ at 30 °C for cleavage of the peptidyl-tRNA ester linkage by the RFC on yeast ribosomes[36,37], raising the question of whether the slow rates we find might be an artifact due to interactions that the dyes attached to the peptide and the tRNA might make with the ribosome. With respect to the effect of Atto 647 labeling of the peptide, we tested this point directly by determining the rate of dissociation at 25 °C of a radioactively labeled peptide devoid of any fluorescent label. Here we employed a Millipore filtration assay in which the cleaved peptide, non-covalently bound to the ribosome, is retained on the filter, while the released peptide is found in the filtrate. As demonstrated in Fig. 4i, the rate constants measured by determining the radioactivity either retained on the filter ($0.30 \pm 0.07$) or found in the filtrate ($0.19 \pm 0.05$) match well the peptide release rates determined in our fluorescent assays (Fig. 4c,g). Given the close similarity in the measured rates of fluorescent Cy3-labeled tRNA$^{Gln}$ dissociation and peptide dissociation, which is mechanistically significant (see Discussion), we think it extremely unlikely that the Cy3 labeling significantly retards tRNA$^{Gln}$ dissociation. This view is buttressed by the much smaller relative

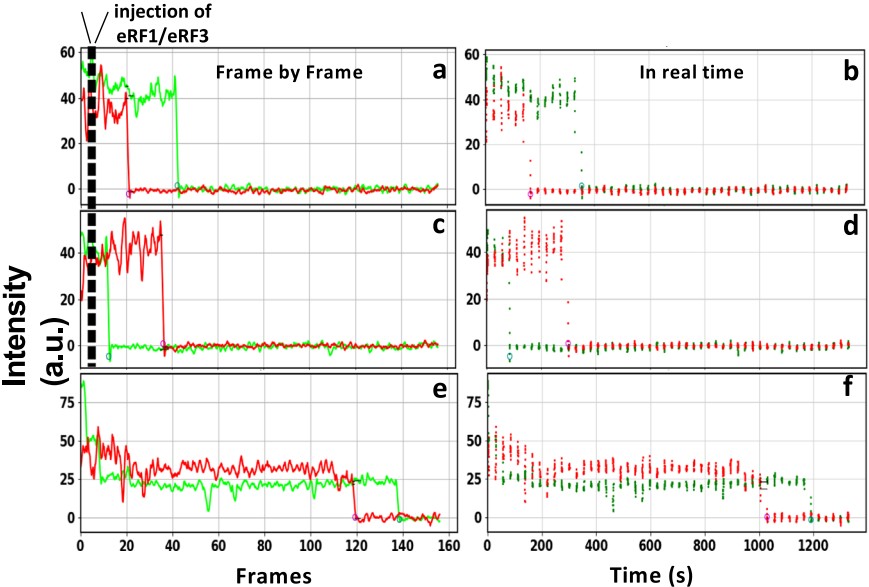

**Fig. 3 Typical traces showing the dissociation of Atto-647 labeled peptide (red) and Cy3-labeled tRNA (green) following eRF1/eRF3 injection.** In the time-lapse experiment the sample was briefly illuminated between fixed time intervals. Sample traces following eRF1/eRF3 injection. In **a** the peptide signal disappears prior to tRNA signal; in (**c**) the tRNA signal disappears prior to peptide signal. **b** and **d** Corresponding real-time scatter plots of the traces presented in (**a**) and (**c**) where each dot represents one frame. **e** Sample trace from a control experiment where only buffer was injected to obtain the photobleaching/spontaneous dissociation rate, which is clearly much slower than the rates seen in (**a**) and (**c**). **f** The real-time scatter plot of the trace presented in (**e**). Source data for (**a**), (**c**), and (**e**) are provided as a Source Data file.

structural change that dye labeling of tRNA represents as compared with dye labeling of a pentapeptide. In other control experiments, we found that addition of ataluren before RFC or simultaneously with RFC gave equivalent results, in both sm-TIRF and ensemble fluorescence anisotropy experiments consistent with the notion that ataluren binds more rapidly to Stop-POST5 complex than does the RFC. In addition, very similar results were obtained in the fluorescence anisotropy experiments with POST5 complexes prepared by combination of isolated 40S and 60S subunits or from 80 S ribosomes salt-treated to facilitate IRES binding, as described in Materials and Methods.

**Ataluren has no direct effects on the rates of peptide and tRNA release, processes which proceed much more slowly than tRNA-peptide bond cleavage.** Cleavage of the tRNA-peptide ester linkage by the RFC has been reported to proceed on yeast ribosomes with rate constants of 5–10 min$^{-1}$ at 30 °C, determined using a thin layer electrophoresis assay[36,37], much faster than the rates of complete release of the two products, ~0.2–0.3 min$^{-1}$, observed here (Fig. 4c, d, g, h). We reasoned that if ataluren inhibits the tRNA-peptide bond cleavage solely by competition with RFC, then the time course of tRNA-peptide hydrolysis might be tracked by adding ataluren at fixed times after RFC addition. Using two syringe pumps triggered at pre-programmed times after camera recording started, Stop-POST5 was treated first with 0.081 μM RFC. Subsequently, 1000 μM ataluren was added to the flow chamber at times varying from 4–25 s. $k_{RF}$ values for peptide and tRNA dissociation are plotted vs. the interval between RFC and ataluren addition in Fig. 4j. Zero time in this plot indicates simultaneous addition of RFC and ataluren. Infinite time indicates no addition of ataluren. The rate constants for elimination of ataluren's inhibitory effect after RFC addition are 10.8 ± 3.6 min$^{-1}$ and 9.6 ± 2.6 min$^{-1}$ for tRNA and peptide respectively, measured at 24 °C. These values are fully compatible with the rate constants reported for peptidyl-cleavage on yeast ribosomes mentioned above[36,37].

**Correlation between individual peptide and tRNA release times.** We next tested whether tRNA and peptide release were linked or independent of one another by calculating the correlation coefficient between peptide and tRNA release times for each ribosome measured in the single molecule experiments. By correlating these times, it was possible to determine whether release of peptide accelerated or retarded tRNA release and, conversely, whether release of tRNA altered the rate of peptide release. Fluorescence excitation laser powers were reduced to 10–20% of that in the other experiments to minimize distortion of any correlation between the release times caused by photo-bleaching, which is not expected to be correlated between the Cy3-tRNA and Atto647-peptide labels. Figure 5 shows scatter plots of individual tRNA lifetimes vs. the corresponding peptide lifetimes for each particle, as well as contour plots of these distributions. Correlation coefficients (CC) between the peptide and tRNA lifetimes were calculated using Eq. (3)

$$CC = \frac{n\sum t_i p_i - \sum t_i \sum p_i}{\sqrt{n\sum t_i^2 - \left(\sum t_i\right)^2}\sqrt{n\sum p_i^2 - \left(\sum p_i\right)^2}} \qquad (3)$$

where $t_i$ and $p_i$ are individual tRNA and peptide lifetimes after RFC addition, respectively, and the summations are conducted over $n$ individual events. CC would equal 1.0 if the two products always dissociated together, fall between 0 and 1 if a preceding slow step limits both events or if dissociation of one product accelerates dissociation of the other, and be <0 if dissociation of one slows dissociation of the other. We found CC to be approximately 0.11 ± 0.03 at saturating RFC and 0.3–0.4 at 16 nM RFC. This value suggests that dissociation times of the two ligands are partly limited by a preceding common kinetic step in the pathway and partly by the two independent (microscopic) dissociation rates. The higher CC at low RFC concentration shows the more prominent effect of the slowed preceding reactions, leading to increased correlation of the peptide and tRNA dissociations. The implications of these results for the kinetic pathway of termination are taken up in Discussion.

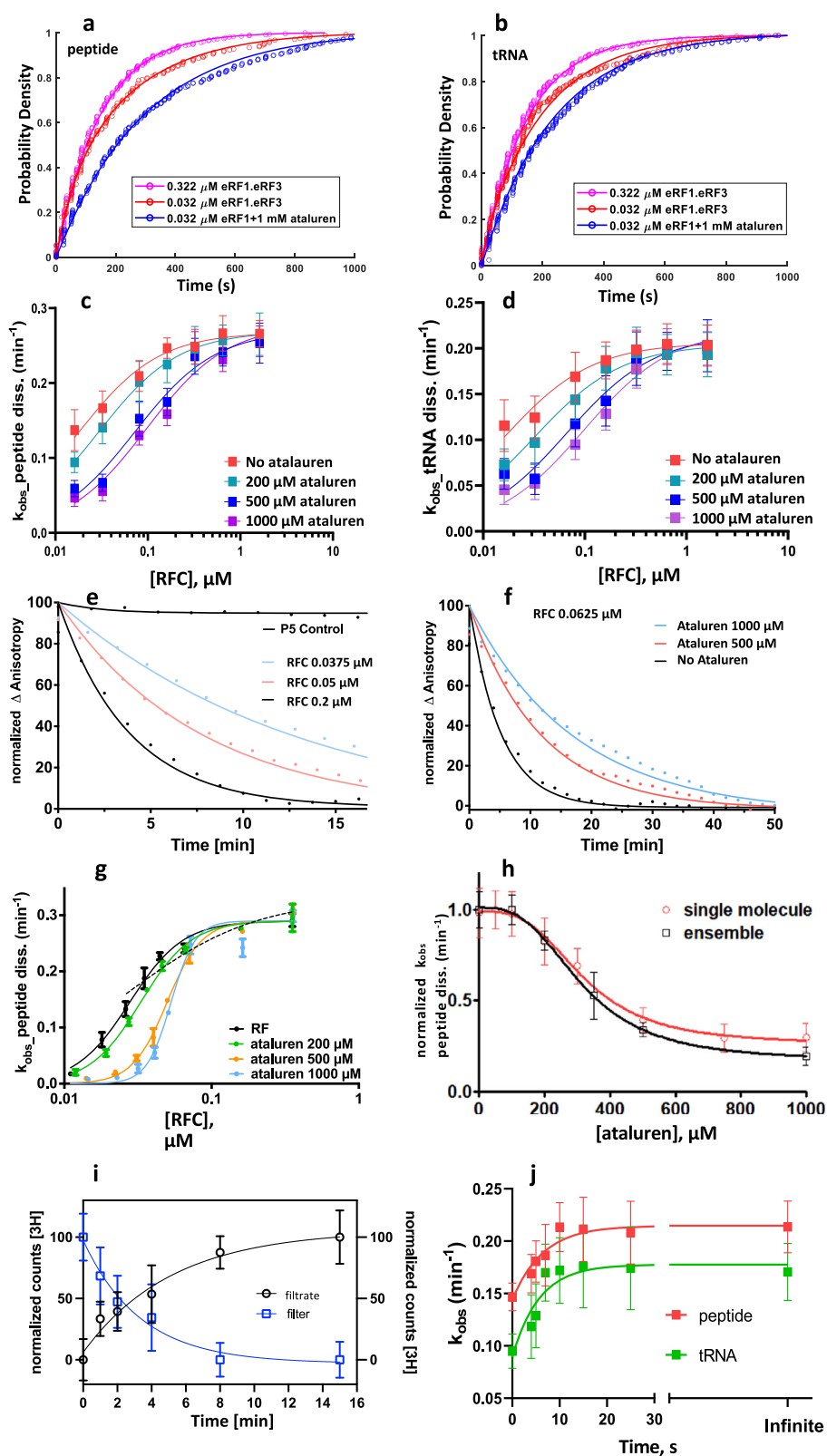

## Discussion

Recently we demonstrated that ataluren stimulation of nonsense codon readthrough results exclusively from its inhibition of RFC-dependent termination of polypeptide synthesis and does so via apparent binding to multiple, probably at least three, sites of the protein synthesis apparatus[27]. Here, using both single molecule and ensemble measurements of release factor activity, we demonstrate that cooperative binding of ataluren to these sites, proceeding with a Hill $n$ of $3.0 \pm 0.6$ and an $EC_{50}$ of $340\,\mu M$, competitively inhibits RFC catalysis of termination (Fig. 4h). We also employ AzAt, a structural (Fig. 1a) and functional (Supplementary Fig. 1)[21] photolabile analog of ataluren, to identify two of

**Fig. 4 Ataluren effects on RFC-dependent termination of polypeptide synthesis.** Cumulative distributions of (**a**) peptide and (**b**) tRNA dissociation times at indicated RFC concentrations, one also with 1 mM ataluren. Each cumulative distribution is constructed from ≥300 kinetic traces. Rates of dissociation of (**c**). peptide and (**d**). tRNA as a function of RFC concentration at different fixed ataluren concentrations. Error bars are ± s.e.m. for $n = 250$–750 trials. Normalized plots of ensemble experiments showing single exponential fits (solid lines) of decimated smoothed raw data (points) of atto647 pentapeptide release reaction measured by fluorescence anisotropy decay vs. time at 25 °C. **e** At indicated RFC concentrations. The control shows the near constancy of observed anisotropy in the absence of added RFC or ataluren. **f** At an RFC concentration of 0.0625 μM and varying ataluren concentrations. **g** The rates of dissociation of atto647 pentapeptide in ensemble experiments as a function of free RFC concentration at varying ataluren concentrations. Error bars are average deviation (a. d.) for $n = 2$–6 independent determinations. Values of n for each point are presented in Supplementary Table 5. **h** Ataluren inhibition of normalized rates of dissociation of atto647 pentapeptide as measured by single molecule (red) and plate reader (black) assays. [eRF1], 32 nM; [eRF3], 0.2 μM and 0.8 μM in the single molecule and ensemble assays, respectively. Error bars are ± s.e.m. $n ≥ 250$ trials (single molecule) and ± a.d., $n = 2$ independent determinations (plate reader). **i** Rates of dissociation of atto647 labeled and unlabeled pentapeptide as measured by millipore filtration, calculated using both filtrate and filter retained values. Error bars are ± s.d., $n = 8$ independent measurements for all points except for the 4 min measurements, for which $n = 11$. [eRF1], 0.2 μM; [eRF3], 0.8 μM; POST5, 0.05 μM. **j** Ataluren inhibition of peptide and tRNA release when added at different times (4–25 s) following RFC addition to Stop-POST5. [RFC], 0.08 μM; [Ataluren], 1 mM. Values at zero-time correspond to simultaneous addition of ataluren and RFC. Error bars are ±s.e.m. for n ≥ 200 trials. Source data for all panels are provided as a Source Data file.

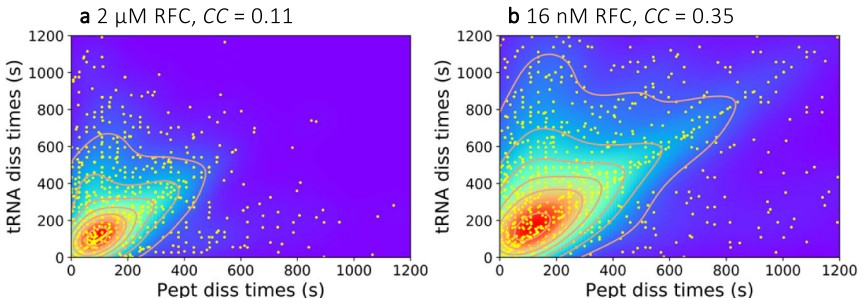

**Fig. 5 Scatter plots of peptide and tRNA dissociation times.** Each point represents dissociation times from an individual ribosome. *CC* is correlation coefficient. **a** 2 μM RFC. **b** 16 nM RFC. Source data are provided as a Source Data file.

these ataluren sites within rRNA by photoincorporation of AzAt. One of these sites, 18S-A1195, is proximal to the ribosome decoding center (Fig. 2g). Both eRF1.eRF3.GDPNP and ataluren inhibit AzAt photoincorporation into this site (Table 1), providing strong evidence that it is a site linked to ataluren inhibition of termination. The second site lies within the PTC (Fig. 2h), where the termination hydrolysis reaction takes place, and is identified by AzAt photoincorporation into nucleotides 26S-A2669, 26S-A2672, and 26S-A3093. This photoincorporation proceeds cooperatively, with a Hill $n$ of $3.3 ± 0.2$ and an EC_50 of 200 μM (Fig. 2e). Compelling evidence that ataluren binding to the PTC is important for inhibition of termination is provided by the similarities in the Hill $n$ and EC_50 values for both ataluren inhibition of termination and AzAt photoincorporation into the PTC, as well as by the inhibition of AzAt photoincorporation by eRF1.eRF3.GDPNP (Table 1).

Two important unknowns remain concerning ataluren inhibition of termination. One is the identity of a likely third ataluren site important for inhibition. The binding site within eRF1 identified by AzAt photoincorporation (Fig. 2b,c) is a possible candidate. Although ataluren binding to this site does not affect $V_{max}$ values for peptide and tRNA departure from the ribosome (Fig. 4c,d,g), it could inhibit termination by weakening RFC binding to the ribosome. However, we currently lack evidence that this is the case. Other possibilities would include a site within a ribosomal protein, which remains to be determined, or an additional site within the RNA fraction of Stop-POST5. For example, a site which had a low yield of AzAt photoincorporation would not have been included within the 22 nucleotides shown in Fig. 2d. In addition, a site within the 22 nucleotides which was excluded from further consideration because of the proximity to eRF1 criterion that we imposed, but which allosterically inhibited eRF1 binding. The second unknown

is the mechanism underlying the cooperative nature of ataluren binding seen in Figs. 2e and 4h. It is reasonable to posit that ataluren binding to a site or sites other than the PTC induces a conformational change within the Stop-POST5 complex which stimulates ataluren binding to the PTC. 18S-A1195 is a plausible candidate for such a site, since AzAt photoincorporation into 18S-A1195 proceeds noncooperatively, and with a lower EC_50 than that seen for photoincorporation into the PTC. Structural studies directed toward determining whether ataluren induces such a structural change, and, if so, through binding to which sites, could elucidate this mechanism.

In cryoelectron microscopy structures of the eRF1.eRF3.GDPNP[38] or eRF1.eRF3.GDPCP[35] bound to pretermination complexes similar to POST-Stop5, eRF1 is bound to eRF3 in a conformation preventing it from extending into the PTC, while interacting strongly with the 18S rRNA decoding center. These structures raise the question of why we detect inhibition by added eRF1.eRF3.GDPNP of AzAt photoincorporation at both sites. This seeming inconsistency is resolved by recent single molecule experiments[37] showing that the binding to a pretermination complex of a modified RFC complex, formed either with a GTPase-deficient eRF3 variant or with GTP-γS in place of GTP, is followed by rapid dissociation (<3 s) of eRF3 from the ribosome, even in the absence of GTP hydrolysis. In our photoincorporation experiments, rapid dissociation of eRF3.GDPNP would allow the eRF1 bound to the ribosome to adopt the elongated conformation that interacts with both the PTC and the decoding center[32,33] during the minute-long photoincorporation process (Supplementary Fig. 12). It remains unclear why eRF3 is still bound to the ribosome in the cryo-EM structures.

Although uncertainties remain in our understanding of the termination process, we think it useful to present a highly simplified and only semi-quantitative kinetic scheme (Fig. 6) for termination of peptide synthesis and its inhibition by ataluren, consistent

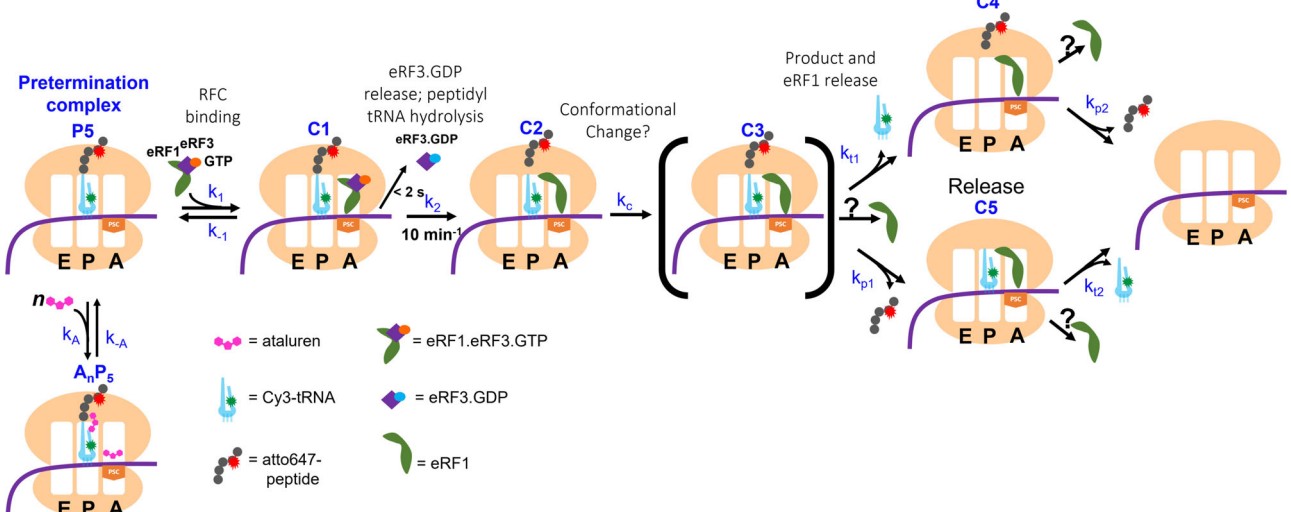

**Fig. 6 A simplified model for RFC catalysis of termination.** This model, consistent with all of our results, as well as with recent published results of others[37], posits that ataluren inhibition results from the cooperative binding of (*n*) molecules of ataluren to the pretermination complex P5 in competition with RFC binding. It also invokes a hypothetical complex C3 resulting from an at least partial rate-determining conformational change, following cleavage of the tRNA-peptide ester bond to account for the similarity in the rate constants of tRNA, peptide, and eRF1 release (see text). The question marks associated with possible eRF1 release are shown to indicate our current uncertainty as to whether eRF1 release is coordinated with either peptide or tRNA release, or proceeds independently of either.

with the results presented above. The results in Fig. 4 show that ataluren is an apparent competitive inhibitor of RFC catalysis of tRNA and peptide release but only acts on steps up to or including the cleavage reaction. This accounts for the placement of ataluren binding prior to formation of complex C1. For simplicity we show *n* molecules of ataluren binding in a single step, and, in the cartoon, indicate two of the binding sites identified by our photoaffinity labeling results as being in the 40S decoding center and the 60 S PTC. In this scheme the relatively rapid cleavage step leading to complex C2 proceeds with a rate constant of $10 \text{ min}^{-1}$. The much slower tRNA and peptide release steps from complex C3 proceed with similar apparent constants of 0.2 to $0.3 \text{ min}^{-1}$ at saturating RFC concentration. Recent results of others under conditions comparable to ours indicate that eRF1 dissociates with a rate constant, $0.2–0.3 \text{ min}^{-1}$ from a yeast ribosome following the cleavage reaction[37], quite similar to the rate constants we measure for peptide and tRNA release. This similarity suggests that all three dissociations proceed via a hypothetical common intermediate (C3, in brackets), possibly involving a conformational change, which is at least partially rate-determining for each dissociation. The presence of C3 in the kinetic scheme is consistent with the positive *CC* value of ~0.1 we have determined, which results in a value of $k_c$ approximately equal to 3 times the value of either $k_{t1}$ or $k_{p1}$ (see Supplementary Notes). We conclude that peptide and tRNA each have their own activation energy barrier for release. Further work will be required to determine how closely, if at all, eRF1 release is coordinated with either peptide or tRNA release.

The pentapeptide release we observe could occur directly from the PTC or via the peptide tunnel, as is the case for much longer peptides. In support of release via the peptide tunnel is the close similarity between our measured rate constant of peptide release ($0.3 \text{ min}^{-1}$, Fig. 4) and that measured at 30 °C for release of a full-length nanoluciferase protein, synthesized in a rabbit reticulocyte lysate, following RFC-dependent termination ($0.35 \text{ min}^{-1}$)[39]. This similarity also provides a strong indication that our results showing that product release is rate-determining for termination is likely to be physiologically relevant, since typical cellular concentrations of both eRF1 and eRF3[40] are considerably in excess of our measured $EC_{50}$ values for RFC of 20–30 nM. Nevertheless,

given the direct coupling of eukaryotic termination with ribosome recycling[41], it will be of interest to determine the effect of adding recycling factors (e.g., Rli1p, ABCE1) on the rates of product release, and experiments to elucidate this point are underway.

The $EC_{50}$ values for ataluren measured in this paper and in our earlier work[21,27] cluster in the range of 150–300 µM, ~10 to 30 times higher than ataluren concentrations employed in the growth media typically used in studies of ataluren stimulation of read-through in intact cells or tissue cultures[42,43]. As we pointed out earlier[27], this difference may be more apparent than real, since ataluren, like other hydrophobic molecules, could be preferentially taken up by cells, such that its cellular concentration far exceeds that present in the cell culture medium[44]. Nevertheless, it would clearly be desirable to develop future TRIDs with greater potency than ataluren for combatting PSC diseases. Elsewhere[27] we have argued that such development should favor TRIDs directed toward inhibiting termination. In that work we showed that even a high concentration (500 µM) of ataluren does not interfere with the binding of near-cognate Trp-tRNA$^{\text{Trp}}$.eE-F1A.GTP ternary complex (TC) to Stop-POST5 during the readthrough elongation cycle. Our present results demonstrate that, in contrast, ataluren binding to both the decoding center and the PTC (Fig. 2g, h) competitively inhibits RFC binding to Stop-POST5 (Fig. 4c, d, g), even though TC and RFC binding sites to pretermination complexes like Stop-POST5 have considerable overlap[35]. While our results are insufficient to precisely locate the ataluren sites within the ribosome, our working hypothesis is that they will lie in areas of the RFC binding site proximal to the decoding center and the PTC which *do not* overlap with the TC binding site. This leads us to suggest that, in the current absence of high resolution structures of ataluren sites, efforts to design TRIDs with higher affinity for pretermination complexes than ataluren should focus on these non-overlapping sites.

## Methods

**Chemicals**. [³H]-AzAt was prepared by catalytic tritiation (Vitrax, Placentia, CA) of 3-[5-(4-azidophenyl)-1,2,4-oxadiazol-3-yl]-6-iodo benzoic acid (6-iodo-AzAt). The chemical purity of the tritiated product was verified by its comigration with authentic AzAt on RP-HPLC analysis (Supplementary Fig. 13). AzAt and 6-iodo-AzAt were

supplied by PTC Therapeutics. Both compounds were characterized by $^1$H NMR (DMSO-d$_6$, 500 MHz): *AzAt*, δ 13.35 (br, s, 1H), 8.6–8.7 (m, 1H), 8.32 (d, 1H, *J* = 7.6 Hz), 8.2–8.3 (m, 2H), 8.1–8.2 (m, 1H), 7.75 (t, 1H, *J* = 7.6 Hz), 7.3–7.5 (m, 2H); *6-iodo-AzAt*, δ ppm 13.67 (br s, 1 H) 8.36 (s, 1 H) 8.16–8.29 (m, 3 H) 7.89 (br d, *J* = 7.32 Hz, 1 H) 7.38 (br d, *J* = 8.24 Hz, 2 H) and mass spectroscopy [*AzAt*, 306.4 [M-H]$^-$, exact mass 307.07; *6-iodo-AzAt*, 432.2 [M-H]$^-$, exact mass 432.97] and each ran as a single peak on RP-HPLC analysis (Supplementary Fig. 13). Atto647-N-hydroxysuccinimide ester (Atto647-NHS) was obtained from Sigma-Aldrich.

**Atto 647-ε-lysine tRNA$^{Lys}$ formation.** Atto647-ε-lysine tRNA$^{Lys}$ was prepared by reacting Atto647-NHS with [$^3$H]Lys-tRNA$^{Lys}$ (*S. cerevisiae*) following a procedure based on that used for the preparation of NBD-ε−[$^3$H]Lys-tRNA$^{Lys}$[45]. Briefly, Atto647 NHS ester dissolved in DMSO (38.5 mM, 10 μl) was added quickly to a 50 mM K$^+$ phosphate buffer, pH 11 (20 μl) and DMSO (50 μl) solution at 25 °C, followed directly by [$^3$H]Lys-tRNA$^{Lys}$ (20 nmol, 165 μM, 120 μl) dissolved in water, reaching a final concentration of 0.1 mM tRNA and 2.0 mM dye at a final pH of 10.8. A color change from dark blue to dark teal was observed. The mixture was stirred in a thermomixer for 4 min at 25 °C, quenched with 4 M acetic acid (17 μl), which restores the original blue color, and directly ethanol precipitated (20% potassium acetate pH 5 (2.7 ml), cold EtOH (17 ml) at −80 °C for 1 h). Ethanol precipitation was repeated twice to remove any residual Atto647-NHS, and the Atto647-ε-lysine tRNA$^{Lys}$ was stored as an aqueous solution containing diethyl pyrocarbonate (0.1%) in small aliquots at −80 °C. The stoichiometry of [$^3$H]-Lys charging was typically 0.35 ± 0.05/tRNA$^{Lys}$. The stoichiometry of Atto647 labeling, measured by UV absorption (ε$_{647}$: 120,000 mol$^{-1}$ cm$^{-1}$), was 0.40 ± 0.05/tRNA$^{Lys}$.

**Ribosome complexes.** 80S.Stop-IRES and Stop-POST5 complexes used in photoaffinity labeling and fluorescence anisotropy experiments were either prepared from shrimp (*Artemia salina*) cyst 40S and 60S subunits, Stop-IRES, aminoacyl-tRNAs (Phe-tRNA$^{Phe}$, Lys-tRNA$^{Lys}$, Val-tRNA$^{Val}$, Arg-tRNA$^{Arg}$, Gln-tRNA$^{Gln}$), yeast elongation factors (eEF1A and eEF2), and GTP (photoaffinity labeling and fluorescence anisotropy experiments), as previously described[46], or via a high KCl treatment of 80 S ribosomes (fluorescence anisotropy experiments). In the latter case, crude 80S ribosomes were dissolved in Buffer 4 (40 mM Tris HCl, pH 7.5, 80 mM NH$_4$Cl, 5 mM Mg(Ac)$_2$,100 mM KOAc, and 3 mM β-mercaptoethanol). and pelleted by microultracentrifugation for 1 h at 540,000 × *g* at 4 °C. The washed 80S ribosomes (2 μM) were combined with Stop-IRES (4 μM) in Buffer 4 supplemented with added 0.4 M KCl and incubated at 37 °C for 30 min. The resulting 80S.Stop-IRES complex was ultracentrifuged through a 1.1 M sucrose solution in Buffer 4 (1.65x of reaction volume) for 1 h at 540,000 × *g* at 4 °C, and the pellet was dissolved in Buffer 4 and stored in small aliquots at −80 °C. This preparation of 80S.Stop-IRES complex was used to make Stop-POST5 complex by a process essentially identical to that used for converting 80 S.Stop-IRES made from 40S and 60S subunits to Stop-POST5 complex[46].

80S.Stop-IRES concentrations were estimated by A$_{260}$ measurement, while the concentration of Stop-POST5 was estimated from both the A$_{260}$ measurement and the stoichiometry of radioactively labeled pentapeptide per 80S ribosome. Typically, 40 ± 5% of the 80S ribosome harbors the pentapeptide. For Atto(pep)-Stop-POST5 and Atto(pep)-Cy-Stop-POST5 preparation, Atto647-εmeasurement$^{Lys}$ replaced lysine-tRNA$^{Lys}$. Also, for Atto(pep)-Cy-Stop-POST5 and Atto(rbsm)-Cy-Stop-POST5 preparation, Gln-tRNA$^{Gln}$(Cy3) replaced Gln-tRNA$^{Gln}$. Finally, for Atto(rbsm)-Cy(tRNA)-Stop-POST5 and Atto(pep)-Cy(ribsm)-Stop-POST5 preparation, ribosomes labeled on r-protein Lys residues to an average stoichiometry of 0.82 Atto647/ribosome and 0.95 Cy3/ribosome, respectively, replaced unlabeled ribosomes. Such labeled ribosomes were prepared using a previously published method for *E. coli* 70 S ribosome labeling with minor modification[47]. In brief, labeling was performed in LB buffer (40 mM HEPES, pH 7.5, 80 mM NH$_4$Cl, 5 mM Mg (Ac)$_2$). A sample of crude 80S shrimp ribosomes, was layered on top of 1.1 M sucrose in LB, then spun at 540,000 x *g* for 90 min. The pellet was washed three times with LB and then dissolved again in LB. For the labeling reaction, a 2-fold molar excess of Atto647N-NHS ester (ATTO-TEC GMBH, dissolved in DMSO) and a 4- fold molar excess of Cy3-NHS ester (Lumiprobe, dissolved in DMSO) was incubated with 5 μM 80 S ribosome in LB at 37 °C for 30 min with stirring. After the reaction, the reaction mixture was layered on the top of 1.1 M sucrose in Buffer 4, then spun at 540,000 × *g* for 90 min. The labeled ribosome pellet was dissolved in Buffer 4. A final spin at 11,952 × g for 2 min removes residual contaminants. The labeling efficiency was calculated by using the molar extinction coefficients of 80S ribosome at 260 nm of Atto647N-NHS ester at 647 nm and of Cy3-NHS ester at 555 nm.

For the single-molecule experiments, Stop-POST5 was prepared from 40S and 60S subunits, and Stop-IRES mRNA covalently attached to biotin at its 3′ end, as described[48], with the following modifications. The oxidation of mRNA was performed in a solution containing mRNA at a concentration of 10–50 A$_{260}$/ml, 100 mM sodium acetate (pH 5.2), and 90 mM sodium m-periodate (prepared fresh). After a 2-hr incubation at room temperature, the periodate was precipitated by adding KCl to a final concentration of 200 mM and incubating for 5 min on ice. The precipitate was removed by centrifugation for 5 min at 10,000 *g*, 2 °C and passage of the supernatant through a Sephadex G-25 column (Nap-5, Pharmacia). EZ-Link hydrazide biotin (ThermoFisher, inc.) was then added to a final concentration of 2 mM from a 50 mM stock in DMSO (prepared fresh).

The biotinylation reaction was carried out for 2 h at room temperature, after which the whole mixture was applied to a Sephadex G-25 column (PD-10, Pharmacia), the high molecular weight fractions were precipitated by ethanol addition, and the resulting pellet pellet was dissolved in DEPC treated H$_2$O. The concentration of biotinylated mRNA was determined by A$_{260}$.

*S. cerevisiae* eEF1A and eEF2 preparations were adapted from the eEF2 preparation method described previously[21,27,46]. Full-length *S. cerevisiae* eRF1 and amino acid residues (166-685) of the eRF3 open reading frame sequences were inserted into the pET-15b (Novagen) plasmid obtained from the laboratory of Allan Jacobson (University of Massachusetts Medical School). eRF1 and eRF3 plasmids were transformed into the *E. coli* BL21(DE3) CodonPlus (Agilent) strain in the presence of ampicillin. eRF1 and eRF3 were isolated from cell lysate as described earlier[21,27,46].

**TRIDs.** Ataluren sodium salt and GJ072 were obtained as gifts from PTC Therapeutics.

**tRNAs.** Yeast tRNA$^{Phe}$ was purchased from Sigma-Aldrich. Other isoacceptor tRNAs were prepared from bulk tRNA (Roche) from either *E. coli* (tRNA$^{Val}$, tRNA$^{Lys}$, tRNA$^{Gln}$), and or *S. cerevisiae* (tRNA$^{Arg}$), via hybridization to immobilized complementary oligo DNAs as described previously[49,50]. *E. coli* and yeast tRNAs were charged with their cognate amino acids as described[21,27,46].

**Photoaffinity labeling with [$^3$H]-AzAt.** eRF1, eRF1.eRF3.GDPNP, Stop-POST5, and 80 S.Stop-IRES samples in Buffer 4, typically 20 μL, were incubated in the dark (20 min for Stop-POST5 and 80S.Stop-IRES complexes, 2 min for release factors) with various concentrations of [$^3$H]-AzAt at 37 °C, transferred to a UV-transparent cuvette which was covered with parafilm to prevent evaporation, and irradiated in a Rayonet photochemical reactor equipped with 300 nm UV lamps for 5 min at room temperature. This was followed by quenching with 2 mM DTT at 37 °C for 5 min under ambient light. The irradiation time gave complete photolysis of the phenyl azide moiety in AzAt (Supplementary Fig. 12). Photolysis was also complete at 5 min in the presence of 1 mM ataluren (Supplementary Fig. 14), ruling out internal filtering effects.

*Stop-POST5 and 80S.Stop-IRES complexes.* 1 μM solutions of either Stop-POST5 or 80S.Stop-IRES, determined by A$_{260}$, were photolabeled by [$^3$H]-AzAt. The sample was then either directly purified by ultracentrifugation through a sucrose cushion to determine the photoincorporation stoichiometry in the complex, or subjected to phenol/chloroform extraction to determine the photoincorporation stoichiometry in the total RNA. To determine the photoincorporation background arising from incomplete removal of noncovalently bound photolyzed AzAt, AzAt was pre-photolysed for 5 min before adding to the complex, followed by the 20 min incubation in dark at 37 °C and subsequent sucrose cushion purification or RNA extraction as described above. This background level, which never exceeded 17 ± 3% of what was observed for the photoincorporation experiment, was subtracted from the observed values in the reported results.

*Photoaffinity labeling of release factors.* Solutions containing eRF1 (10 μM), eRF3 (20 μM), and 1 mM GMPPMP were pre-incubated at 37 °C for 2 min to form the ternary complex prior to photolabeling with [$^3$H]-AzAt. After quenching, reaction mixtures were loaded onto a 4–15% mini protean SDS-PAGE gel for protein separation and isolation. Gel slices containing eRF1 or eRF3, as shown by Coomassie Blue R250 staining were crushed and the radioactivity was extracted in extraction buffer (50 mM Tris-HCl, 150 mM NaCl, and 0.1 mM EDTA; pH 7.5) by vigorously shaking at 70 °C for 30 min. The data fitting was performed using the software Prism. Labeling results of eRF1 alone and within the eRF1.eRF3.GDPNP complex were fit with a one-site binding model and a two-site binding model, respectively.

*RNA-Seq* procedures employed closely follow those published[29,51] except as noted. RNA-seq libraries were prepared for PAL, PRE, UV, and NUL samples, as defined in Results, of Stop-POST5 and 80S.IRES complexes, using published procedures[29,51] with the slight modifications that a 12% 0.13 M Tris/45 mM Borate/2.5 mM EDTA (pH 7.6)- Urea PAGE gel replaced a 10% PAGE gel and the selected RNA fragment size was 40–70 bp rather than 60–70 bp. In brief, RNA was extracted and randomly fragmented by ZnCl$_2$ (Sigma). After separation on the TBE-Urea PAGE gel, RNA fragments, visualized by ethidium bromide staining (used for all RNA gels) were sliced and extracted from the gel by crushing the gel and vigorously shaking at 70 °C in 400 μL of water for 15 min. The 3′ ends were dephosphorylated by T4 Polynucleotide Kinase (New England Biolab). The miRNA cloning linker 2 (5′App/CACTCGGGCACCAAGGA/3′ddC, Integrated DNA Technologies) was used as the universal 3′ linker and pre-adenylated using the 5′ DNA Adenylation Kit (New England Biolab) with the provided protocol. Reverse transcription, self-circularization, and PCR amplification were performed essentially the same as previously described[51]. Library sequencing was performed with Illumina NextSeq with single-end mode for up to 75 bp read length.

Sequencing reads were processed by Cutadapt 2.10 (https://cutadapt.readthedocs.io/en/stable/) to remove 3′ end linkers and then read into Shapemapper2[52] to count the mutations and effective sequencing reads, using the STAR Aligner mode[53] The mutation rate of each nucleotide was calculated by dividing the number of the reads containing the mutations by the total number of reads, outputted as N.

For the PAL sample, the mutation rate fold change was calculated by dividing the mutation rate of the PAL sample by the mutation rate of the PRE sample. The average mutation count was calculated by averaging the number of mutations of the PAL and PRE samples. MA plots (Supplementary Fig. 6) were displayed by plotting the mutation rate fold change of each nucleotide as a function of the average mutation count of each nucleotide. To determine possibly significant sites of AzAt labeling, the dataset was first screened for points that meet the criteria of a mutation rate fold change ≥1.5 and an average mutation count ≥100. Points meeting these criteria were further screened with Z-factor > 0 and p-value < 0.05 to determine statistical significance. The Z-factor (z) was calculated using Eq. (4).

$$z = 1 - \frac{1.96 \times (\sigma PAL + \sigma PRE)}{|\triangle N|} \qquad (4)$$

$\sigma$ is the square root of the mutation rate (M) divided by the read depth (C) and further normalized by the average mutation rate (A): $\sigma = \sqrt{(M/C)}/A$. $\triangle N$ is the difference in the normalized mutation rate ($N = M/A$) between the PAL and PRE samples.

To determine the impact of UV treatment, the mutation rate fold change was calculated by dividing the mutation rate of the UV sample by the mutation rate of the NUL sample. The delta mutation rate (UV-NUL) was also calculated as an extra layer of filtering. Nucleotides having >10 UV/NUL, > 0.01 (UV – NUL), and >100 average mutation counts were selected, and identified by sequence homology within the yeast rRNA sequence (Supplementary Table 3). Native rRNA modifications within the Stop-POST5 (NUL) and 80S.IRES (NUL) samples having ≥0.2 average mutation rate were similarly identified within the yeast rRNA sequence (Supplementary Table 2).

**RNaseH Fragment Assay.** RNA was prepared from the [3H]-AzAt photoaffinity labeled ribosome by phenol-chloroform extraction. RNA: DNA duplexes were formed by incubating total RNA (0.2 μM) with an oligonucleotide pair (1 μM of each, sequences shown Supplementary Table 4), flanking a specific RNA region, in 22.8 μL of water at 95 °C for 2 min, followed by annealing at 50 °C for 30 s and quenching on ice. Six units (1.2 μL) of RNase H (NEB) and 6 μl of 5x RNase H Reaction Buffer were next added to the RNA: DNA duplex and incubated at 37 °C for 1 hour, followed by phenol-chloroform extraction of RNA. The digested RNA was separated on a 12% TBE-Urea PAGE gel, and the target band along with the adjacent upper and bottom gels were sliced by a 2 mm gel cutter. The gel slices were crushed and the [3H] labeled rRNA fragment was extracted in water by vigorously shaking at 70 °C for 30 min, or by overnight extraction in elution buffer (0.3 M sodium acetate, 0.25% SDS, 1 mM EDTA). The slurry was removed by centrifugation and the eluted [3H]-labeled rRNA in the supernatant was quantified. The stoichiometry of AzAt photoincorporation was calculated based on a gel recovery yield of 20 ± 2%, determined using an authentic sample of [3H]-Gln-tRNA$^{Gln}$.

**Plate reader anisotropy assay.** Plate reader assays were run in a Tecan F200® 96-well plate reader (Greiner® black, non-binding, chimney flat) equipped with 640 nm excitation and 700 nm polarization filters using 96-well plates. Atto(pep)-Stop-POST5 in Buffer 4 (0.1 μM, 80 μl) containing GTP (1 mM) was added to plate wells ± ataluren at 25 °C. eRF1 and eRF3 aliquots stored at −80 °C were thawed for 15 s at 37 °C and added directly to a premade ice-cold GTP (1 mM) ± ataluren solution in Buffer 4. The resulting release factor solution was incubated at 25 °C for 1 min, and then added quickly in 80 μL portions to the Atto(pep)-Stop-POST5 containing wells with a multichannel pipette to the Atto(pep)-Stop-Post5 wells. The final concentrations were: eRF1, 0.025–0.4 μM; eRF3, 0.8 μM; Atto(pep)-Stop-Post5, 0.05 μM. Fluorescence anisotropy decay graphs were fit with GraphPad Prism, using the one phase decay model to obtain peptide dissociation rates. All measurements shown in Fig. 4g, h were independently repeated between 2 and 6 times (average 3.3 times) with error ranges determined as average deviations. The number of independent observations for each curve in Fig. 4g, h varied between 16–34.

**Millipore filtration assay.** Stop-POST5 complex containing [3H]-labeled peptide (0.1 μM, 10 ul) in Buffer 4 containing 1 mM GTP was prepared separately for each time point. Release factor mixture in Buffer 4 (eRF1, 0.4 μM; eRF3, 1.6 μM; GTP 1 mM) was prepared and kept on ice. A nitrocellulose filter (MF-Millipore, 0.45 μM, MCE Membrane, 25 mm diameter, hydrophilic) was wetted with Buffer 4 for at least 15 seconds and placed on a fritted vacuum filter funnel with a scintillation vial in the vacuum flask for filtrate collection. Stop-POST5 and RF vials were warmed for 1 min at 25 °C and mixed (for 0 time point Buffer 4 was used), giving the following final concentrations: Stop-POST5, 0.05 μM; eRF1, 0.2 μM; eRF3, 0.8 μM. At indicated times, aliquots were applied to the filter, and the filter was immediately washed with Buffer 4 (200 μl). The filtrate and wash were collected in a scintillation vial. The filter was placed in a separate vial and shaken with ethyl acetate (1 ml) for 1 h. The fritted vacuum filter funnel was washed three times with Buffer 4 (200 μl) into the filtrate/wash scintillation vial. Scintillation fluid was added to the vials (4 ml), shaken vigorously and radioactivity was determined.

**Preparation of PEG-passivated slides.** PEG-passivated slides were prepared according to previously published procedures with minor modifications[54]. In brief, slides and coverslips were sonicated at 40 °C in the order of acetone (10 min), methanol (10 min), 200 mM KOH (20 min), and ethanol (10 min). Cleaned slides and coverslips were treated in a fume hood with 1 ml 3-aminopropyltriethoxysilane, 5 ml acetic acid, in 94 ml methanol at room temperature overnight, sealed with parafilm, and then incubated with polyethylene glycol (PEG, Laysan Bio, Inc., containing 20% (w/w) mPEG succinimidyl valerate, MW 2000 and 1% biotin-PEG-SC, MW 2000) in 0.1 M sodium bicarbonate (pH 8.3) for 4 h. Slides and coverslips were then washed with Milli-Q water, dried by clean $N_2$, placed in 50 ml Falcon tubes, vacuum-sealed under $N_2$ in food saver bags, and stored at −20 °C.

**Flow chamber construction.** The flow chambers for fast injection of reaction mixtures were made as described previously[55]. The sample flow chambers (8 μL) were formed on slides with holes drilled using a 1.25 mm diamond-tipped drill bit. Polyethylene tubing with 0.97 mm outer diameter (Warner Instruments) was inserted into each hole, sealed with 5 min epoxy, and trimmed flush. Double-sided tape laid between the tubes served as spacers and separated the flow chambers. The coverslips were then sealed in place via the double-sided tape and epoxy at their edges.

**Immobilization of ribosome Stop-POST 5 complexes.** PEGylated flow chambers were incubated in 0.5 mg/ml streptavidin (Sigma-Aldrich) for 5 mins and washed with Buffer 4. Biotinylated Stop-POST5 containing FK(Atto647)VRQ-tRNA$^{Gln}$ (Cy3) in the P-site, was formed at room temperature by incubating Stop-POST4, containing FK(Atto647)VR-tRNA$^{Arg}$ in the P-ste and an empty A-site, eEF1A, GTP, and Gln-(Cy3)tRNA$^{Gln}$ for 5 min and then injected into the streptavidin-coated slide chamber. After a 5 min incubation, excess unbound ribosome was flowed out. Then eRF1/eRF3 ± TRIDs was injected dynamically at 3 s while recording. The injection dead time was subtracted for each particle.

**Single-molecule fluorescence imaging.** All sm-TIRF studies were carried out at 24 °C. All dilutions, complex formation, and single-molecule imaging were carried out in Buffer 4 with an added enzymatic oxygen scavenging system of 2 mM protocatechuic acid (PCA), 50 nM protocatechunate 3,4 dioxygenase (PCD, Sigma Aldrich), 1 mM cyclooctatetraene (COT, Sigma Aldrich), 1 mM 4-nitrobenzyl alcohol (NBA, Sigma Aldrich), and 1.5 mM 6-hydroxy-2,5,7,8-tetramethyl-chromane-2-carboxylic acid (Trolox, Sigma-Aldrich)[56].

Image stacks were recorded at 100 ms frame rate on a custom-built objective-type total internal reflection fluorescence (TIRF) microscope based on a commercial inverted microscope (Eclipse Ti-E, Nikon) and capable of performing alternating-laser excitation (ALEX) between 532 nm and 640 nm laser beams using an acousto-optic tunable filter (AOTF[55]) to switch wavelengths. For the time-lapse experiment, timing of the movies, programmed into NIS-Elements (Nikon) microscope software, were recorded by illuminating the sample for 50–20 frames at 10 frames per s with 15 s or 24 s intervening dark intervals. Video recording was started and then eRF1/eRF3 ± TRIDs was injected from a triggered syringe pump at 3 s during the recording. The injection dead time was subtracted for each particle. Triggering of the shutters, AOTF, and pump was synchronized using LabView scripts driven from the camera exposure signal.

**Data analysis.** Collected movies were analyzed by a custom-made software program developed as an ImageJ plugin (http://rsb.info.nih.gov/ij)[57], and further analyzed using Python. Distributions of peptide and tRNA dissociation times were fit to Eq. (5), using maximum likelihood estimation[58,59], where P is the probability density.

$$P = k_1 e^{-k_1 t} \qquad (5)$$

Data and fitted curves were plotted as cumulative probability densities. In Fig. 4c, d, h, j the number of kinetic traces in each distribution ranged from 200–750, and the number of independent duplicates ranged from 2–3. Error bars are the mean ± s.e.m which is calculated from the total number of traces.

**Reporting summary.** Further information on research design is available in the Nature Research Reporting Summary linked to this article.

## Data availability
The raw sequencing reads fastq data files used in this study are available in the BioProject database under accession code ID PRJNA714648. Raw sequencing reads in fastq format are available as Supplementary Data 1. All other Source data are provided as Source Data file.

## Code availability
The source code programs for single molecule analysis are stored in a Google drive with the following link: https://drive.google.com/drive/folders/1SJnqYL5mWw5O0RrGG_K6L49ztOu03Zjh?usp=sharing

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

## Acknowledgements

We gratefully acknowledge many helpful discussions with Boris Zynshteyn on the use of RNA-Seq, and gifts of GJ072, ataluren sodium salt and 3-[5-(4-azidophenyl)-1,2,4-oxa-diazol-3-yl]-6-iodo benzoic acid from PTC Therapeutics and of the eERF1 and eRF3 plasmids from Allan Jacobson. This work was supported by research grants to B.S.C. (PTC Therapeutics; Cystic Fibrosis Foundation-COOPER21G0; and NIH GM127374), to Y.E.G. (NIH GM118139), to DMC (NIH GM118510) and to grant S10OD021573 (Y.E.G., PI) for the PicoQuant MicroTime 200 instrument.

## Author contributions

S.H. was responsible for planning, executing, and interpreting the results of the photo-affinity labeling experiments; A.B. and C.F. were responsible for planning, executing, and interpreting the results of the single-molecule experiments; M.D.G. was responsible for the preparation of Atto 647-ε-lysine tRNA$^{Lys}$ and for planning, executing, and interpreting the results of the plate reader anisotropy and Millipore filtration assays; H.L. was responsible for the preparation and characterization of ribosomes, tRNAs, CrPV-IRES-mRNA, and both elongation and termination, factors, DMC made valuble suggestions for the conduct of the RNA-Seq experiments and provided financial support, Y.E.G. was responsible for overseeing the single-molecule experiments, writing the manuscript, and providing financial support, B.S.C. was responsible for overseeing the photoaffinity labeling, plate reader anisotropy, and Millipore filtration experiments, writing the manuscript, and providing financial support.

## Competing interests

The authors declare no competing interests.
