## [Peer Review File · Nature Communications]

Ataluren Binds to Multiple Protein Synthesis Apparatus Sites and Competitively Inhibits Release Factor Dependent TerminationEditorial Note: This manuscript has been previously reviewed at another journal that is not operating a transparent peer review scheme. This document only contains reviewer comments and rebuttal letters for versions considered at *Nature Communications*.

REVIEWERS' COMMENTS

Reviewer #1 (Remarks to the Author):

The authors did an excellent job replying to the comments from the first review. However, there are still questions about some of the interpretation of results.

Comments:

1) The authors did a nice job explaining how they settled in on the ataluren binding sites for study to 3 sites. However, it cannot be disregarded that ataluren was shown to bind to 22 sites. Indeed, these results suggest that while ataluren is shown here to inhibit translation termination in this reductionist assay, it does so in a rather non-specific manner, as only 13.6% of the binding sites contribute to translation termination activity.

2) The authors do a nice job of highlighting studies where ataluren has somewhat succeeded in promoting PTC readthrough. However, ataluren is far from a success in the clinic and there are several studies that show it has no significant effect in vivo. The authors should do the literature and clinical trials justice in ataluren's failures as well. Further, the manuscript would benefit from a section that describes how individual PTCs (position, sequence context, etc) could be more or less difficult to readthrough by TRIDS. This discussion is common in most PTC suppression studies. Do the same rules apply to TRIDS as they do for aminoglycosides for instance?

Reviewer #2 (Remarks to the Author):

It appears that many errors were caught during this peer review process. The authors have now addressed at least to some degree of satisfaction all of the comments and suggestions of both Referees of this work in their revised manuscript

Response to referee #1:

Please address the remaining concerns of referee #1:

1. The authors did a nice job explaining how they settled in on the ataluren binding sites for study to 3 sites. However, it cannot be disregarded that ataluren was shown to bind to 22 sites. Indeed, these results suggest that while ataluren is shown here to inhibit translation termination in this reductionist assay, it does so in a rather non-specific manner, as only 13.6% of the binding sites contribute to translation termination activity.

Non-site-specific labeling, including from the reactive photoproduct formed from the photoaffinity label in solution, is always present in PAL experiments. This is especially true in the present case because the relatively low affinity binding of AzAt to its target sites necessitated using a high concentration of AzAt in the experiments presented in Figs. 2d and 2e. This is why the results in Tables 1 and S1 demonstrating the site-specificity of the labeling of RNase H Fragments I-III and the lack of specificity in the labeling of RNase H fragments IV and V are so important. We do not consider the relatively low 13.6% labeling of the site-specifically labeled sites as problematic, because the large majority of the remaining 86.4% labeling, spread over the entire ribosomal RNA, is likely to come either from very weakly binding sites, none of which are anywhere close to eRF1, or from solution.

2) The authors do a nice job of highlighting studies where ataluren has somewhat succeeded in promoting PTC readthrough. However, ataluren is far from a success in the clinic and there are several studies that show it has no significant effect in vivo. The authors should do the literature and clinical trials justice in ataluren's failures as well. Further, the manuscript would benefit from a section that describes how individual PTCs (position, sequence context, etc) could be more or less difficult to readthrough by TRIDS. This discussion is common in most PTC suppression studies. Do the same rules apply to TRIDS as they do for aminoglycosides for instance?

In response to the comment regarding literature and clinical trials, we have added significant text to paragraph 1 of the Introduction, (lines 35-37, 41-46) which also includes seven new literature references (#s 4-6, 9, 16-18). We do not think it necessary to add a section on how position and sequence context of a nonsense codon affect readthrough, as these variables were not investigated in this manuscript.